# Occurrence of Aflatoxin M$_1$ in Milk and Dairy Products Traded in São Paulo, Brazil: An Update

**Carlos Humberto Corassin** [1], **Aline Borowsky** [2], **Sher Ali** [1], **Roice Eliana Rosim** [1] **and Carlos Augusto Fernandes de Oliveira** [1,*]

[1] Faculty of Animal Science and Food Engineering, University of São Paulo, Pirassununga 13635-900, SP, Brazil
[2] Luiz de Queiroz College of Agriculture, University of São Paulo, Piracicaba 13635-900, SP, Brazil
[*] Correspondence: carlosaf@usp.br; Tel.: +55-019-3565-4145

**Abstract:** The aim of this study was to conduct an up-to-date investigation on the occurrence levels of aflatoxin M$_1$ (AFM$_1$) in samples of raw milk ($n$ = 40), pasteurized milk ($n$ = 44), ultra-high temperature (UHT) milk ($n$ = 27), Minas cheese ($n$ = 57), and yogurt ($n$ = 44) traded in São Paulo state, Brazil. AFM$_1$ was extracted from fluid milks and dairy products using immunoaffinity columns and determined by high performance liquid chromatography. AFM$_1$ was detected at the mean level of 0.080 ± 0.071 μg/L or kg in 72 samples (34.0%) evaluated in the study ($n$ = 212). Detectable levels of AFM$_1$ were observed in five samples of raw milk (12.5%), 16 samples of pasteurized milk (36.4%), 13 samples of UHT milk (48.1%), 27 samples of cheese (47.4%), and 11 samples of yogurt (25.0%), although none of them had concentrations above the maximum permitted levels (MPL) for AFM$_1$ adopted in Brazil. However, 11.7% ($n$ = 13) of samples of raw, pasteurized, and UHT milks would have AFM$_1$ concentrations above the MPL of 0.05 μg/L adopted in the EU. The maximum level was detected in one cheese sample containing 0.695 μg/kg. Although none of the samples exceeded the Brazilian MPL, the high frequencies of AFM$_1$ in Brazilian milk products warrant concern about their contribution to the human exposure to aflatoxins. Because aflatoxins are among the most potent carcinogens known, the results of this trial stress the need for stringent measures in the milk production system to avoid AFM$_1$ in milk and derived products.

**Keywords:** AFM$_1$; analysis; milk products

---

## 1. Introduction

Mycotoxins are important contaminants produced as secondary metabolites by certain toxigenic fungi during their growth on food and feedstuffs, which can develop several toxic effects such as hepatotoxicity, nephrotoxicity, neurotoxicity, teratogenicity, immunosuppression, mutagenicity, and carcinogenicity, among others [1–3]. Aflatoxins are produced by fungal species from *Aspergillus* genus, especially *A. flavus*, *A. parasiticus*, and *A. nomius* [1]. As broadly dispersed, these fungi species and their harmful metabolites are a major problem to the food and feed industries [1]. Up to now, more than 18 compounds have been classified as "aflatoxin—AF" [2,3]. In food and feed materials, the highly toxic compound that normally occurs under any natural conditions is the aflatoxin B$_1$ (AFB$_1$), followed by G$_1$ (AFG$_1$), B$_2$ (AFB$_2$), and G$_2$ (AFG$_2$). After the ingestion of AFB$_1$ by animals, the hepatic biochemical transformation of the molecule produces the hydroxylated derivative known as aflatoxin M$_1$ (AFM$_1$), which is excreted into milk and other body fluids [4,5]. Both AFB$_1$ and AFM$_1$ are classified as group 1 human carcinogens by the International Agency for Research on Cancer (IARC) [6–9]. Many countries still lack a proper system to avoid food contamination with aflatoxins, especially milk and derived products that have been shown to be risky for consumption [10]. Milk is largely consumed by humans, therefore, the AFM$_1$ level in milk is constantly regulated worldwide [11,12]. The amount of AFM$_1$ excreted in milk of lactating animals has been expressed as a percentage of the AFB$_1$

intake from contaminated feed. The European Food Safety Authority has estimated that the transfer rate from $AFB_1$ in feed to $AFM_1$ in cow's milk is 1–2% on average, although in high-productivity cattle, it can increase to 6% [13].

As extensively indicated in the literature, milk products such as pasteurized milk and ultra-high temperature (UHT), if not controlled properly, may contain $AFM_1$, thus posing a significant risk to consumers [3,4,9,10]. In addition, $AFM_1$ binds to casein and remains bound to it, even in the production processes of dairy products including powdered milk, cheese, and yogurt [14–16]. In line with this tight binding capacity and the high heat resistance of the toxin, $AFM_1$ in dairy products cannot be completely removed by standard heat-based pasteurization or sterilization processes [14,17]. Thus, studies have demonstrated that human exposure to $AFM_1$ can be associated with the consumption of the given products [14,18–20]. For instance, human exposure to an exceeded limit of $AFM_1$ in any of these products can contribute as causative agents for human diseases such as hepatocellular carcinoma (HCC) [21]. Thus, determination of $AFM_1$ in milk and dairy products are essential for control purposes, which requires appropriate analytical techniques such as enzyme-linked immunosorbent assay (ELISA) [22,23], high performance liquid chromatography (HPLC) [24], and HPLC coupled tandem mass spectrometry (MS/MS) [25].

$AFM_1$ levels in yogurt, pasteurized, and UHT milk, along with related health factors have been recently described in important milk-producing countries including China [26] and Brazil [5]. High frequencies (59–83%) of $AFM_1$ in yogurts and pasteurized and UHT milks exceeding the maximum permitted level (MPL) of 0.05 µg/L of the European Union (EU) were reported in China [26]. Moreover, about 79% of pasteurized and UHT milk were found to be contaminated with $AFM_1$, with high seasonal variation, whereas a high $AFM_1$ prevalence was observed in the winter [27]. A recent study confirmed that higher risk concerns regarding $AFM_1$ contamination of milk in Croatia were associated with autumn and winter [23]. However, a survey conducted in Bangladesh indicated no or small significant differences correlated to the seasonality [22]. In Brazil, previous studies regarding the occurrence of $AFM_1$ showed higher incidence rates from 63% to 100%, and levels ranging from 0.0002 to 0.106 µg/L among diverse milk products [28–30]. According to a Brazilian study [18], UHT milk, powdered milk, and infant formula on evaluation have exhibited 0.150–1.02 µg/kg of $AFM_1$, whose maximum level surpassed the MPL of 0.5 µg/L adopted in Brazil [11,12]. Recently, raw milk used in the manufacture of Minas cheese were evaluated in cheese processing plants from the Brazilian state of São Paulo, and the results indicated that 39 and 29% of samples were contaminated with $AFM_1$ at mean levels of 0.028 ± 0.009 µg/L and 0.113 ± 0.092 µg/kg, respectively [31]. However, there is little information available on the recent occurrence data regarding $AFM_1$ in pasteurized milk, UHT milk, Minas cheese, and yogurts commercially available in retail markets for human consumption in Brazil. Therefore, the objective of this study was to conduct an up-to-date investigation on the occurrence levels of $AFM_1$ in raw milk collected in dairy processing plants and in dairy products traded in São Paulo state, Brazil.

## 2. Materials and Methods

### 2.1. Sampling Design

A total of 212 samples of raw milk ($n = 40$), pasteurized milk ($n = 44$), UHT milk ($n = 27$), Minas cheese ($n = 57$), and yogurt ($n = 44$) were collected between January and August of 2022. Samples of raw milk were collected in 10 dairy plants located in the northeast region of São Paulo state, Brazil, while samples of pasteurized milk, UHT milk, cheese, and yogurt were collected in supermarkets from the same region. In each location, duplicate samples of raw milk (500 mL), processed milks (pasteurized and UHT: 1-L original bottle), Minas cheeses (500-g original package), and yogurts (900-g original bottle) were collected and transported to the laboratory in a cooler with dry ice and stored at 4 °C until the analysis of $AFM_1$.

### 2.2. Reagents and Instruments

Analytical grade reagents were used in all determinations of AFM1 in the milk and dairy samples. Ultra-pure water was provided by a Milli-Q$^®$ system (Millipore, Bedford, MA, USA). The $AFM_1$ standard was purchased from Sigma Chemical Company (St Louis, MO, USA). Methanol and acetonitrile of HPLC grade were obtained from EM Science (Gibbstown, NJ, USA).

All chromatographic analyses were conducted on a Shimadzu$^®$ (Kyoto, Japan) 10VP high performance liquid chromatographic (HPLC) system equipped with a Shimadzu$^®$ (Kyoto, Japan) 10 AXL fluorescence detector (excitation at 360 nm and emission above 440 nm). The chromatographic column used was a Kinetex $C_{18}$ column (Phenomenex, Torrance, CA, USA) $4.6 \times 150$ mm with 2.6 μm particle size, coupled to a pre-column with a 0.5 μm filter. An isocratic mobile phase composed by methanol:water:acetonitrile (61.4:28.1:10.5, *v/v/v*) was used in all chromatographic runs, with a flow rate of 0.50 mL/min.

### 2.3. Sample Preparation

$AFM_1$ was extracted from samples of milk and dairy products using immunoaffinity columns (Aflatest$^®$, Vicam, Watertown, MA, USA) with minor modifications as described and validated in the laboratory by Jager et al. [32]. For raw, pasteurized, and UHT milks, 40 mL aliquots of each sample were transferred to Falcon tubes and pre-heated at 37 °C for 10 min. Then, 1 g of NaCl was added to the tube and centrifuged (SOLAB SL-700, Piracicaba, Brazil) at 3500 rpm for 5 min. After separating the upper fat layer, the sample was submitted to a second centrifugation step to remove the remaining fat. Samples were filtered through Whatman no. 4 filter paper, and 30 mL-aliquots were passed into immunoaffinity columns attached to a manifold (Supelco, Bellefonte, PA, USA) and vacuum pump (Primatec, Itu, Brazil) at a flow rate of 2–3 mL/min. Twenty mL of ultra-pure water (Milli-Q$^®$, Millipore, Bedford, MA, USA) was used to wash the column before eluting $AFM_1$ with 1 mL of methanol. The eluate was recovered in glass vials and submitted to evaporation under nitrogen to near dryness. The final extract was resuspended in 1 mL of methanol/water (50:50, *v/v*), and 25 μL was injected into the HPLC system, as described in Section 2.2. The retention time for $AFM_1$ was 5.9 min.

$AFM_1$ in yogurt and Minas cheese was determined by weighing 8 g of ground cheese in a Falcon tube and then adding 2 g of sodium chloride, 22 mL of methanol, and 13 mL of water. The extraction was conducted by mixing the mixture in a mixer for 1 min, followed by centrifugation at 3500 rpm for 10 min. The upper liquid phase was filtered through Whatman no. 4 filter paper, and 20 mL of the filtrate was diluted with 40 mL of ultra-pure water (Milli-Q$^®$, Millipore, Bedford, MA, USA). The mixture was passed through the immunoaffinity column (2–3 mL/min) attached to a manifold (Supelco, Bellefonte, PA, USA) and vacuum pump (Primatec, Itu, Brazil), and the subsequent extraction steps were identical as described for the milk samples.

Calibration curves to quantify $AFM_1$ in the milk and milk products (yogurt and cheese) were prepared using $AFM_1$ standard solutions (Sigma$^®$) prepared in acetonitrile at levels ranging from 0.1 to 1 μg/L and 0.1 to 20 μg/L, respectively. The $AFM_1$ concentrations in the samples were linearly correlated with the integrated peak areas of the standards. The limits of detection (LOD) and quantification (LOQ) were determined for each analytical method considering a signal-to-noise ratio of 3:1 and 10:1, respectively. Linearity was verified in all chromatographic runs by checking the coefficient of determination ($r^2$) through visual observation of the residual plots of the calibration curves.

The performance of the method for $AFM_1$ in the milk and dairy products samples has been presented elsewhere [32], describing LOD and LOQ values of 0.0025 and 0.0080 μg/L for $AFM_1$ in fluid milk, respectively. LOD and LOQ of $AFM_1$ in the cheese and yogurt samples were 0.017 and 0.055 μg/L, respectively.

## 3. Results and Discussion

The occurrence levels of AFM$_1$ in samples of raw milk collected at dairy plants and the processed milks and dairy products commercially available in São Paulo state are summarized in Table 1. Throughout the analyses, AFM$_1$ was detected in 72 samples (34.0%) of milk and dairy products evaluated in the study (*N* = 212). Detectable levels of AFM$_1$ were observed in five samples of raw milk (12.5%; *n* = 40), 16 samples of pasteurized milk (36.4%; *n* = 44), 13 samples of UHT milk (48.1%; *n* = 27), 27 samples of Minas cheese (47.4%; *n* = 57), and 11 samples of yogurt (25.0%; *n* = 44). The maximum level was detected in one cheese sample containing 0.695 µg/kg. These results indicate a high incidence of low levels of AFM$_1$ in milk and dairy products, especially in fluid milks (pasteurized and UHT) and in cheese.

**Table 1.** Aflatoxin M$_1$ (AFM$_1$) levels in raw milk collected at dairy plants, and pasteurized milk, UHT milk, cheese, and yogurt commercially available in São Paulo, Brazil.

| Product | <LOD [1] | | LOD–0.050 [2] | | 0.051–0.250 [2] | | 0.251–0.500 [2] | | 0.501–1.00 [2] | | Mean [3] | Range [4] |
|---|---|---|---|---|---|---|---|---|---|---|---|---|
| | *n* | % | *n* | % | *n* | % | *n* | % | *n* | % | | |
| Raw milk (*n* = 40) | 35 | 87.5 | 1 | 2.5 | 4 | 10.0 | 0 | 0 | 0 | 0 | 0.114 ± 0.070 | 0.014–0.182 |
| Pasteurized milk (*n* = 44) | 28 | 63.6 | 10 | 22.7 | 6 | 13.7 | 0 | 0 | 0 | 0 | 0.032 ± 0.033 | 0.003–0.117 |
| UHT milk (*n* = 27) | 14 | 51.9 | 10 | 37.0 | 3 | 11.1 | 0 | 0 | 0 | 0 | 0.080 ± 0.038 | 0.020–0.148 |
| Minas cheese (*n* = 57) | 30 | 52.6 | 16 | 28.1 | 7 | 12.3 | 1 | 1.7 | 3 | 5.3 | 0.122 ± 0.190 | 0.017–0.695 |
| Yogurt (*n* = 44) | 33 | 75.0 | 5 | 11.4 | 6 | 13.6 | 0 | 0 | 0 | 0 | 0.050 ± 0.025 | 0.017–0.091 |
| Total (*N* = 212) | 140 | 66.0 | 42 | 19.8 | 26 | 12.3 | 1 | 0.5 | 3 | 1.4 | 0.080 ± 0.071 | 0.003–0.695 |

The header spanning columns reads: **Number of Samples According to the Level of AFM$_1$**

[1] LOD: Limit of detection of the analytical methods: 0.0025 µg/L (raw, pasteurized and UHT milks) and 0.017 µg/kg (cheese and yogurt). [2] µg/L or µg/kg. [3] Values (µg/L or µg/kg) are reported as mean ± standard deviation, for samples analyzed in duplicate containing detectable levels of AFM$_1$. [4] Minimum and maximum levels of AFM$_1$ (µg/L or µg/kg). UHT: Ultra high temperature.

Toxigenic fungal molds are widely disseminated, and inherently affect the food and feed chain. In particular, milk and further dairy products are globally consumed by people of all ages (e.g., toddlers to the elderly), thus unacceptably higher frequencies and levels of AFM$_1$ in milk products can pose risks to human health. In this study, the raw, pasteurized, and UHT milks showed the occurrence of AFM$_1$ at the mean levels of 0.114 ± 0.070, 0.032 ± 0.033, and 0.080 ± 0.038 µg/L, respectively. All samples had AFM$_1$ levels below the Brazilian MPL of 0.5 µg/L [12], which is in contrast to the higher occurrence levels of AFM$_1$ in milk from several countries [26,33–36]. Compared with the data reported here, much higher AFM$_1$ concentrations in the pasteurized milks at maximum levels of 4.81, 2.33, and 1.14 µg/L were determined in Pakistan [36], India [34], and Ethiopia [35], respectively. Additionally, in contrast with our results, UHT milks with higher AFM$_1$ concentrations of 1.54 [36], 2.58 [34], and 0.56 µg/L [18] were previously reported in Brazil. In these studies, AFM$_1$ concentrations in the pasteurized or UHT milks exceeded the established EU (0.05 µg/L) or the Brazilian limits (0.5 µg/L) [8]. In the same way, higher incidence rates (63–100%) of AFM$_1$ at levels ranging from 0.0002 to 0.106 µg/L were observed in milk products traded in Brazil [28–30]. Thus, compared with previous data on the frequencies and levels of AFM$_1$ in milk and dairy products, our results indicate that control measures regarding the occurrence of AFB$_1$ in feed for dairy cows have been improved in Brazil. However, 13 samples (34%) of fluid milk analyzed (raw, pasteurized and UHT) in our study had AFM$_1$ levels above the MPL of 0.05 µg/L adopted in the EU [11]. Moreover, the mean concentration of AFM$_1$ in yogurt samples was 0.050 ± 0.025 µg/L, and none of the samples analyzed exceed any established limits for raw milk used for the manufacture of dairy products (0.5 µg/L of Brazil or 0.05 µg/L of the EU), which corroborates the low occurrence levels of AFM$_1$ in this type of milk product. Much higher mean concentrations of AFM$_1$ at 1.63, 0.40, and 0.09 µg/L have been described in yogurt samples from Ethiopia [35], Yemen [33], and Lebanon [37], respectively. However, AFM$_1$ concentrations or its variabilities in milk products from different countries can indicate

geographical or climatic effects, poor-quality control, and unsupervised feed and dairy products, respectively.

Minas cheese is another largely consumed, traditional Brazilian dairy product that has been studied regarding AFM$_1$ contamination [20,38]. The mean level of AFM$_1$ in cheese samples collected from retail markets in our study was 0.122 ± 0.190 µg/L, although only three samples (5%) showed a moderate concentration range between 0.50 and 1.00 µg/kg, which was below the normal limit of 2.5 µg/kg settled in Brazil [12]. This is consistent with the frequency of positive samples (29%) and the mean concentration of AFM$_1$ (0.113 ± 0.092 µg/kg) in the Minas cheese collected in cheese processing plants from São Paulo [31]. However, studies over the past decades have demonstrated higher AFM$_1$ levels in the stated samples. For example, 74.7% cheese samples demonstrated AFM$_1$ at the levels of 0.02 to 6.92 µg/g [39], hence indicating some samples with non-compliance levels when compared to the MPL of 2.5 µg/L adopted for cheeses by the Brazilian regulations [12,40]. In agreement with the data reported in this study, lower levels ranging from 0.050 to 0.31 µg/g were observed in Iranian cheeses, which were related to variabilities in the environmental or seasonal conditions [38].

The lack of awareness regarding the AFB$_1$ contamination of feed for dairy cattle and limitations in analytical services remains the most important causes of the high incidence of AFM$_1$ in dairy products in Brazil. In addition, there are no regulations regarding MPL for aflatoxins in feed for dairy cattle in Brazil. Franco et al. [41] found AFB$_1$ at levels ranging from 8.7 to 390 µg/g in 13% of the 45 feed samples collected in Brazilian small-scale farms. However, Brazilian milk production has undergone several modernization processes over the past decades, leading to significant improvements in the overall quality of milk produced in the country [35], which may have contributed to the low levels of AFM$_1$ within Brazilian MPL values for milk and dairy products, as determined in the present study.

## 4. Conclusions

High incidences (12.5–47.4%) of low levels of AFM$_1$ (mean: 0.080 ± 0.071 µg/L or kg) were observed in raw milk collected from dairy plants as well as in pasteurized milk, UHT milk, yogurt, and cheeses commercially available in São Paulo state, Brazil. Although these levels were below the limits established by Brazilian regulations (0.5 µg/L), 34% of the samples of raw, pasteurized, and UHT milks had AFM$_1$ concentrations above the MPL of 0.05 µg/L adopted in the EU. The frequent occurrence of AFM$_1$ in Brazilian milk products warrants concern about their contribution for the human exposure to dietary aflatoxins, taking into account their hepatocarcinogenity. Hence, the results of this trial indicate that stringent measures such as good agricultural practices and the adequate storage of dairy cow feed are needed to avoid AFM$_1$ in milk and derived products. Further studies should be carried out to evaluate the occurrence levels of AFM1 in other dairy products such as milk powder and other types of cheese available for human consumption in Brazil.

**Author Contributions:** Conceptualization, C.H.C.; Methodology, C.H.C., R.E.R. and C.A.F.d.O.; Validation, R.E.R.; Formal Analysis, S.A.; Investigation, A.B., S.A. and R.E.R.; Resources, C.H.C. and C.A.F.d.O.; Data curators, C.H.C. and C.A.F.d.O.; Writing—Original Draft Preparation, C.H.C., A.B. and S.A.; Writing—Review & Editing, C.A.F.d.O.; Supervision, C.H.C.; Project Administration, C.H.C. and C.A.F.d.O.; Funding Acquisition, C.H.C. All authors have read and agreed to the published version of the manuscript.

**Funding:** This research was funded by the São Paulo Research of Foundation, FAPESP (grant number 2022/03952-1 and 2017/20081-6), and by the Brazilian National Council for Scientific and Technological Development, CNPq (Grant #314419/2021-7 and #304262/2021-8).

**Institutional Review Board Statement:** Not applicable.

**Data Availability Statement:** Data are contained within the article.

**Acknowledgments:** The authors highly acknowledge FAPESP and CNPq for their financial support.

Conflicts of Interest: The authors declare no conflict of interest.

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
