# Peer review of "Occurrence of Aflatoxin M1 in Milk and Dairy Products Traded in São Paulo, Brazil: An Update"

_2624-862X, doi:10.3390/dairy3040057_

Round 1

Reviewer 1 Report

This paper reports the current occurrence of aflatoxin M1 (AFM1) in milk (raw, pasteurized, UHT) and dairy product (yoghurt, cheese) samples from São Paulo state (Brazil). The raw milk samples (n=40) were collected in dairy plants, while samples of pasteurized milk (n=44), UHT milk (n=27), yogurt (n=44) and cheese (n=57) were collected in supermarkets, between January and August of 2022.

Due to its toxicity, the presence of AFM1 in milk and dairy products is considered undesirable, and due to the regular consumption of dairy products, its presence in these products represents a serious public health concern. Therefore, the prevalence and amount of AFM1 in milk and dairy products need to be constantly measured and planned in the food chain to minimize it to provide the health of consumers. In this way the approach of the study is interesting due to the problem addressed and the data provided but a thorough review should be conducted regarding the way it is written.

The introduction has an adequate background, but the objective is not clearly stated. The analytical method employed is appropriate, however, some aspects of its performance need to be clarified. Regarding experimental design, further information about the samples collected in supermarkets would be desirable. Specifically, the type of cheeses analyzed should be defined in detail. It is well known that the level of AFM1 in cheese can vary according to different factors, in addition to the presence of toxin in milk, including the type of cheese and the ripening time, among others.

The discussion is written in a confusing way and is poor in establishing association between the results obtained and other published works about the prevalence and levels of AFM1 in milk and dairy products analyzed in various regions of the world.

Specific comments

Lines-43-44 I suggest the authors to see the paper / scientific opinion and information reported in the EFSA Journal 2004, 39, 1–27. (. https://doi.org/10.2903/j.efsa.2004.39). EFSA has estimated that the transfer rate from aflatoxin B1 in feed to aflatoxin M1 in cow’s milk is 1–2% on average, although in high-productivity cattle it can increase to 6%.

Lines 62-64 The results shown in these lines should be considered in the discussion section

Lines 69-70 “the results indicated that a large number of these products were contaminated with AFM1”. Please, be more concise and provide specific details. In that case, the results shown could be also considered in the discussion section.

Lines 72-75 “Therefore, the objective of this study was to conduct an up-to-date investigation on the occurrence levels of AFM1 in milk and dairy products from dairy processing plants in São Paulo state, Brazil”. Is this really the aim of the study? Were the collected samples produced in dairy processing plants located in São Paulo? In this case, it should be specified in the sampling design section.

Line 91- “Aflatest WB Immunoaffinity columns” Please check if it is correct

Lines 129-132 (Method’s performance) The sample preparation procedure in yogurt has undergone important modifications with respect to the original method proposed by Jager et al (2013). In the mentioned work, LOD and LOQ were 0.017 and 0.055 μg/L, respectively. After procedure modification, do the LOD and LOQ values remain the same in yogurt?

Lines 156-172- The authors show the occurrence of AFM1 in the milk (raw, pasteurized, UHT) and yogurt samples as mean ± standard deviation levels. However, they compare their results with the maximum values reported by other authors. This comparison is not appropriate. On the other hand, the maximum values obtained in this study are missing.

Line 166- I suggest authors include the reference “European Commission. Commission Regulation (EC) No 1881/2006, setting maximum levels for certain contaminants in foodstuffs. Off. J. Eur. Union 2006, 364, 5–24”

Lines 169-170- In the European Union (EU) maximum levels have been set for aflatoxin M1 in raw milk, heat-treated milk and milk for the manufacture of milk-based products (0,050 μg/kg), as well as for infant formulae and follow-on formulae, including infant milk and follow-on milk, and dietary foods for special medical purposes intended specifically for infants (0,025 μg/kg) (Regulation (EC) No. 1881/2006). However, no limits have been established for aflatoxin M1 in yogurt.

Lines 175-176- This sentence is confusing. It does not refer to cheese in general, but to a specific type of cheese. Does it intend to define the type of cheese used in this study? As mentioned above, this should be described in the material and methods section.

Author Response

Reviewer #1:

This paper reports the current occurrence of aflatoxin M1 (AFM1) in milk (raw, pasteurized, UHT) and dairy product (yoghurt, cheese) samples from São Paulo state (Brazil). The raw milk samples (n=40) were collected in dairy plants, while samples of pasteurized milk (n=44), UHT milk (n=27), yogurt (n=44) and cheese (n=57) were collected in supermarkets, between January and August of 2022.

Due to its toxicity, the presence of AFM1 in milk and dairy products is considered undesirable, and due to the regular consumption of dairy products, its presence in these products represents a serious public health concern. Therefore, the prevalence and amount of AFM1 in milk and dairy products need to be constantly measured and planned in the food chain to minimize it to provide the health of consumers. In this way the approach of the study is interesting due to the problem addressed and the data provided but a thorough review should be conducted regarding the way it is written.

Answer: Thanks for the comments. The manuscript was thoroughly revised, as suggested.

The introduction has an adequate background, but the objective is not clearly stated. The analytical method employed is appropriate, however, some aspects of its performance need to be clarified. Regarding experimental design, further information about the samples collected in supermarkets would be desirable. Specifically, the type of cheeses analyzed should be defined in detail. It is well known that the level of AFM1 in cheese can vary according to different factors, in addition to the presence of toxin in milk, including the type of cheese and the ripening time, among others.

Answer: As detailed in the responses below, the objectives of the study and the methodological concerns pointed out were revised. In fact, only Minas cheese were evaluated in the study, so this information was amended throughout the manuscript.

The discussion is written in a confusing way and is poor in establishing association between the results obtained and other published works about the prevalence and levels of AFM1 in milk and dairy products analyzed in various regions of the world.

Answer: The discussion was improved, as requested.

Specific comments

Lines-43-44 I suggest the authors to see the paper / scientific opinion and information reported in the EFSA Journal 2004, 39, 1–27. (. https://doi.org/10.2903/j.efsa.2004.39). EFSA has estimated that the transfer rate from aflatoxin B1 in feed to aflatoxin M1 in cow’s milk is 1–2% on average, although in high-productivity cattle it can increase to 6%.

Answer: Thanks for the useful reference, which was included in the revised manuscript.

Lines 62-64 The results shown in these lines should be considered in the discussion section

Answer: The data was amended in the discussion section, as suggested

Lines 69-70 “the results indicated that a large number of these products were contaminated with AFM1”. Please, be more concise and provide specific details. In that case, the results shown could be also considered in the discussion section.

Answer: Specific details were included, and the data were considered in the Discussion section.

Lines 72-75 “Therefore, the objective of this study was to conduct an up-to-date investigation on the occurrence levels of AFM1 in milk and dairy products from dairy processing plants in São Paulo state, Brazil”. Is this really the aim of the study? Were the collected samples produced in dairy processing plants located in São Paulo? In this case, it should be specified in the sampling design section.

Answer: The aims of the study were re-phrased for clarity.

Line 91- “Aflatest WB Immunoaffinity columns” Please check if it is correct

Answer: This sentence was excluded, as the immunoaffinity columns are mentioned further in section 2.3.

Lines 129-132 (Method’s performance) The sample preparation procedure in yogurt has undergone important modifications with respect to the original method proposed by Jager et al (2013). In the mentioned work, LOD and LOQ were 0.017 and 0.055 μg/L, respectively. After procedure modification, do the LOD and LOQ values remain the same in yogurt?

Answer: The analytical method for AFM1 in yogurt was the same as described for cheese, not for milk as wrongly mentioned in the original manuscript. This was corrected in the revised manuscript. So, the analytical procedures and attributes of the method including LOD and LOQ for AFM1 in yogurt samples were the same as described by Jager et al. (2013).

Lines 156-172- The authors show the occurrence of AFM1 in the milk (raw, pasteurized, UHT) and yogurt samples as mean ± standard deviation levels. However, they compare their results with the maximum values reported by other authors. This comparison is not appropriate. On the other hand, the maximum values obtained in this study are missing.

Answer: Although the maximum levels in milk products, except cheeses, are not displayed in the manuscript, they can be inferred from the range values as presented in Table 1, which are useful for comparisons with data previously reported in the literature. In addition, when appropriate, some comparisons are made with mean values reported in previous works.

Line 166- I suggest authors include the reference “European Commission. Commission Regulation (EC) No 1881/2006, setting maximum levels for certain contaminants in foodstuffs. Off. J. Eur. Union 2006, 364, 5–24”

Answer: Thanks for the suggestion, the reference was included in the manuscript.

Lines 169-170- In the European Union (EU) maximum levels have been set for aflatoxin M1 in raw milk, heat-treated milk and milk for the manufacture of milk-based products (0,050 μg/kg), as well as for infant formulae and follow-on formulae, including infant milk and follow-on milk, and dietary foods for special medical purposes intended specifically for infants (0,025 μg/kg) (Regulation (EC) No. 1881/2006). However, no limits have been established for aflatoxin M1 in yogurt.

Answer: Thanks for the clarification. This sentence was corrected, as required.

Lines 175-176- This sentence is confusing. It does not refer to cheese in general, but to a specific type of cheese. Does it intend to define the type of cheese used in this study? As mentioned above, this should be described in the material and methods section.

Answer: The sentence was re-phrased for clarity. In fact, Minas cheese was the focus of the study, so this information was amended throughout the manuscript.

Reviewer 2 Report

please see the attached file.

Author Response

Reviewer #2:

(Comments annotated in the manuscript file)

-L.15: I think a small letter n is what is needed (as its for sample).

Answer: The “N” was replaced with “n”, as suggested.

-L.22: May be here can say the % that exceeded the EU limit.

Answer: The information required was amended in the revised manuscript.

-L.23: Can say "Although none of the samples exceeded permitted levels, the high ----".

Answer: Done.

-L.78: See earlier comment.

Answer: Done: The “N” was replaced with “n”.

-L.141-142: But says earlier they were within the acceptable levels.

Answer: This sentence was re-phrased for clarity.

-L.162: Possible explanation?

Answer: A possible explanation was amended in the revised manuscript.

-L.186: Perhaps can highlight something on AFB1 contamination in feeds.

Answer: A sentence regarding the occurrence of AFB1 in feed in Brazil was amended, as suggested.

-L.187: May be improved quality of feeds has led to reduced contamination (all samples were within the required limits/ local).

Answer: This sentence was adjusted, as suggested.

-L.198: Not clear - does this mean they had?

Answer: Yes, they had, as amended in the revised manuscript.

Reviewer 3 Report

The authors examined the objective investigation on the occurrence levels of AFM1 in milk and dairy products from dairy processing plants in São Paulo state, Brazil. They reported that high incidences of low levels of AFM1 were observed in raw milk collected from dairy plants, as well as in pasteurized milk, UHT milk, yogurt, and cheeses commercially available in São Paulo state, Brazil. The manuscript is well organized. The presented paper is interesting, but the following corrections should be done.

Below are my concerns and suggestions to improve the manuscript,

1. The abstract is missing a piece of brief information on the effects of mycotoxins. 

2. Please provide more details about the mycotoxin effects in the introduction.

3. In particular, aflatoxin M1 is a potent toxic mycotoxin that is classified as a Group 1 human carcinogen by the International Agency for Research on Cancer (IARC).  Please add references with IARC.

4. The conclusion needs to be highly quantitative and should be discussed in more detail.

5. I recommend adding the following current references related to the mycotoxin in the introduction of this manuscript to improve its updated,

https://doi.org/10.1016/j.tifs.2021.01.093

https://doi.org/10.3390/toxins13070440

https://doi.org/10.3390/chemosensors9120363

https://doi.org/10.3390/foods11131959

https://doi.org/10.1016/j.foodchem.2022.134302

https://doi.org/10.1016/j.talanta.2021.122779

6. The detection methods of AFM1 should be added to the introduction section.

7. In the whole paper, the authors should use the abbreviation once it was defined. There are repeat abbreviations. Please check lines 125 and 130.

8. In the conclusions, some advances, current issues, and future scope should also be described.

9.  Please provide more details about the detection time and the sample volume. The authors must explain the advantages of the developed technique. 

10. The preparation process images of dairy samples can be added to the manuscript.

Author Response

Reviewer #3:

The authors examined the objective investigation on the occurrence levels of AFM1 in milk and dairy products from dairy processing plants in São Paulo state, Brazil. They reported that high incidences of low levels of AFM1 were observed in raw milk collected from dairy plants, as well as in pasteurized milk, UHT milk, yogurt, and cheeses commercially available in São Paulo state, Brazil. The manuscript is well organized. The presented paper is interesting, but the following corrections should be done.

Answer: Thanks for the comments. The manuscript was revised accordingly.

Below are my concerns and suggestions to improve the manuscript,

  1. The abstract is missing a piece of brief information on the effects of mycotoxins.

Answer: Due to word count limitations in the Abstract, a very brief information on the harmful effects of aflatoxins was amended, as requested.

  1. Please provide more details about the mycotoxin effects in the introduction.

Answer: Done.

  1. In particular, aflatoxin M1 is a potent toxic mycotoxin that is classified as a Group 1 human carcinogen by the International Agency for Research on Cancer (IARC). Please add references with IARC.

Answer: IARC references were included in the manuscript, as suggested.

  1. The conclusion needs to be highly quantitative and should be discussed in more detail.

Answer: Quantitative data were amended in the concluding statements, when appropriate.

  1. I recommend adding the following current references related to the mycotoxin in the introduction of this manuscript to improve its updated,

https://doi.org/10.1016/j.tifs.2021.01.093

https://doi.org/10.3390/toxins13070440

https://doi.org/10.3390/chemosensors9120363

https://doi.org/10.3390/foods11131959

https://doi.org/10.1016/j.foodchem.2022.134302

https://doi.org/10.1016/j.talanta.2021.122779

Answer: Thanks for the suggestions. The appropriate references directly linked with the scope of the study (focused on aflatoxin M1 in milk products) were amended in the Introduction section of the revised manuscript.

  1. The detection methods of AFM1 should be added to the introduction section.

Answer: Done.

  1. In the whole paper, the authors should use the abbreviation once it was defined. There are repeat abbreviations. Please check lines 125 and 130.

Answer: Checked.

  1. In the conclusions, some advances, current issues, and future scope should also be described.

Answer: The information required was included in the Conclusion section.

  1. Please provide more details about the detection time and the sample volume. The authors must explain the advantages of the developed technique.

Answer: The retention time of AFM1 in the chromatographic run was amended in the revised manuscript. Sample volume injected in the HPLC system was already mentioned in the original manuscript. As explained in section 2.3, the analytical method was validated in our laboratory, so it was not developed in the present study.

  1. The preparation process images of dairy samples can be added to the manuscript.

Answer: The authors thank the suggestion, but unfortunately there is no process images of dairy samples in this study. 

Round 2

Reviewer 1 Report

Dear Authors

The manuscript has been greatly improved compared to its original version. Most of the changes suggested by the reviewers have been introduced to  the revised manuscript. However, I have a couple of comments:

About the answer: “Although the maximum levels in milk products, except cheeses, are not displayed in the manuscript, they can be inferred from the range values as presented in Table 1…”. Sorry, I don't agree with this statement. The maximum levels cannot be properly inferred from the data presented in Table 1. It would be much more understandable for the reader to add a column with the range of contamination (minimum - maximum)

Lines 200-202. There is a mistake in this sentence. The indicated values 4.76, 0.89 and 0.55 μg/L are not mean concentrations, but maximum values of aflatoxin M1 in yogurt samples from Ethiopia, Yemen and Lebanon respectively. Please review the data and write the correct mean concentrations values.

Author Response

Answers to Reviewers:

We have addressed the additional comments and suggestions of the Reviewers regarding the manuscript and included below a point-by-point response to the Reviewers. In addition, the changes done in the new, revised manuscript were highlighted in yellow.

Reviewer #1:

The manuscript has been greatly improved compared to its original version. Most of the changes suggested by the reviewers have been introduced to the revised manuscript. However, I have a couple of comments:

About the answer: “Although the maximum levels in milk products, except cheeses, are not displayed in the manuscript, they can be inferred from the range values as presented in Table 1…”. Sorry, I don't agree with this statement. The maximum levels cannot be properly inferred from the data presented in Table 1. It would be much more understandable for the reader to add a column with the range of contamination (minimum - maximum)

Answer: An additional column indicating the range of contamination (minimum – maximum) was amended in Table 1, as requested. In addition, the mean ± SD values of AFM1 in the present study were corrected in Table 1 and throughout the text, as they were re-calculated considering only the detectable levels in positive samples (previously, the means were calculated wrongly using AFM1 levels above and below LOD). We apologize for this mistake and understand that these changes do not modify the discussion of the data presented.

Lines 200-202. There is a mistake in this sentence. The indicated values 4.76, 0.89 and 0.55 μg/L are not mean concentrations, but maximum values of aflatoxin M1 in yogurt samples from Ethiopia, Yemen and Lebanon respectively. Please review the data and write the correct mean concentrations values.

Answer: The correct mean concentrations reported in those studies were amended in the revised manuscript.

Reviewer 3 Report

Authors have carefully checked and modified this manuscript. Now it can be accepted for publication in this journal without further revision.

Author Response

Thanks for the comments.